# Epigenetic Regulation of Neutrophils in ARDS

**DOI:** 10.3390/cells14151151

**Published:** 2025-07-25

**Authors:** Jordan E. Williams, Zannatul Mauya, Virginia Walkup, Shaquria Adderley, Colin Evans, Kiesha Wilson

**Affiliations:** 1Department of Pathology, Microbiology & Immunology, University of South Carolina School of Medicine, Columbia, SC 29209, USA; jordan.williams@uscmed.sc.edu (J.E.W.); zannatul.mauya@uscmed.sc.edu (Z.M.); virginia.walkup@uscmed.sc.edu (V.W.); 2Department of Pharmacology, Physiology, and Neuroscience, University of South Carolina School of Medicine, Columbia, SC 29209, USA; shaquria.adderley@uscmed.sc.edu; 3Department of Cell Biology & Anatomy, University of South Carolina School of Medicine, Columbia, SC 29209, USA

**Keywords:** ARDS, neutrophils, epigenetics, lncRNA, miRNA, histone modification

## Abstract

Acute respiratory distress syndrome (ARDS) is an inflammatory pulmonary condition that remains at alarming rates of fatality, with neutrophils playing a vital role in its pathogenesis. Beyond their classical antimicrobial functions, neutrophils contribute to pulmonary injury via the release of reactive oxygen species, proteolytic enzymes, and neutrophil extracellular traps (NETs). To identify targets for treatment, it was found that epigenetic mechanisms, including histone modifications, hypomethylation, hypermethylation, and non-coding RNAs, regulate neutrophil phenotypic plasticity, survival, and inflammatory potential. It has been identified that neutrophils in ARDS patients exhibit abnormal methylation patterns and are associated with altered gene expression and prolonged neutrophil activation, thereby contributing to sustained inflammation. Histone citrullination, particularly via PAD4, facilitates NETosis, while histone acetylation status modulates chromatin accessibility and inflammatory gene expression. MicroRNAs have also been shown to regulate neutrophil activity, with miR-223 and miR-146a potentially being biomarkers and therapeutic targets. Neutrophil heterogeneity, as evidenced by distinct subsets such as low-density neutrophils (LDNs), varies across ARDS etiologies, including COVID-19. Single-cell RNA sequencing analyses, including the use of trajectory analysis, have revealed transcriptionally distinct neutrophil clusters with differential activation states. These studies support the use of epigenetic inhibitors, including PAD4, HDAC, and DNMT modulators, in therapeutic intervention. While the field has been enlightened with new findings, challenges in translational application remain an issue due to species differences, lack of stratification tools, and heterogeneity in ARDS presentation. This review describes how targeting neutrophil epigenetic regulators could help regulate hyperinflammation, making epigenetic modulation a promising area for precision therapeutics in ARDS.

## 1. Introduction

Acute respiratory distress syndrome (ARDS) is an illness characterized by increased vascular permeability, hypoxemia, and hyperinflammation in the lungs [1]. The inflammatory response is central to the pathogenesis of ARDS; however, it is not the sole pathological mechanism of this disease [2]. ARDS leads to dysregulation of the immune response and contributes to impaired gas exchange and pulmonary edema [1]. The cytokine storm associated with ARDS exacerbates epithelial and endothelial damage, which ultimately decreases lung function [1]. Neutrophil infiltration and the formation of neutrophil extracellular traps (NETs) promote oxidative stress and proteolytic degradation through neutrophil elastase and matrix metalloproteinases (MMPs) [3]. Infections, pneumonia, sepsis, trauma, and oral inhalation of harmful substances are initiators of ARDS [2]. These insults drive the cascade of immune dysregulation, causing capillary disruption, which impairs the transport of O_2_ and CO_2_.

Chemokine and cytokine gradients drive the migration and activation of neutrophils. Chemotactic factors that drive neutrophil migration include interleukin-8 (IL-8), tumor necrosis factor (TNF-α), monocyte-inhibitory protein 2 (MIP-2), C-X-C motif chemokine 5 (CXCL5), platelet-activation factor (PAF), and N-formylmethionyl-leucyl phenylalanine (fMLP) [4,5]. The inflammatory response in ARDS is also intricately linked to neutrophil heterogeneity [5,6]. Recent studies suggest that neutrophils can be classified into different phenotypes. N1 neutrophils exhibit pro-inflammatory properties, whereas N2 neutrophils demonstrate anti-inflammatory characteristics [7]. The balance between these pro- and anti-inflammatory neutrophil profiles is crucial in determining the clinical trajectory of ARDS in affected patients [8].

Histone modification, non-coding RNAs, and DNA methylation are involved in the epigenetic regulation of neutrophils. Studies have shown that variation during granulopoiesis via DNA methylation impacted neutrophil maturation and activation, further highlighting epigenetics in fine tuning the immune responses. [9]. Histone modifications, such as citrullination and acetylation, regulate chromatin structure and gene accessibility, influencing NET formation [10,11]. These modifications can function in the resolution of inflammation, by balancing apoptosis and cell survival signaling pathways to control excessive tissue damage [11]. Non-coding RNAs, such as microRNAs and long non-coding RNAs, modulate neutrophil responses through post-transcriptional regulation, thereby influencing neutrophil plasticity [12]. Therapeutic inventions have the potential to arise through understanding the epigenetic mechanisms of neutrophils in ARDS.

This review aims to integrate new findings on epigenetic regulation, as well as the recently discovered heterogeneity of neutrophils in ARDS, into therapeutic strategies and personalized medicine. Acknowledging the epigenetic mechanisms unique to different neutrophil subsets, we highlight new opportunities for precision treatments that balance the emphasis on robust inhibition of pro-inflammatory, damaging phenotypes while maintaining homeostatic phenotypes needed for pathogen clearance and phagocytosis. Information collected regarding epigenetic biomarkers of disease can also be used as predictive markers and guide physicians’ treatment strategies.

## 2. Epigenetic Mechanisms

### 2.1. Overview of Epigenetic Mechanisms

Epigenetic mechanisms cause heritable changes in chromosomes without altering the underlying DNA sequence. These modifications influence gene accessibility and expression by modifying chromatin structure, allowing cells to adapt gene expression to environmental stimuli [13]. Key epigenetic modifications discussed in this manuscript as potential therapeutic targets include DNA methylation, histone modifications, and regulatory non-coding RNAs (ncRNAs), which determine cellular identity, function, and response to stress or inflammation [13].

### 2.2. DNA Methylation Mechanisms

DNA methylation is characterized by the addition of methyl groups to the fifth carbon of cytosine residues to produce 5-methyl-cytosine [14]. The addition of a methyl group alters the physical characteristics of DNA, leading to the inhibition of proteins from recognizing the DNA or allowing other proteins to bind to it [15]. This epigenetic modification prevents transcription factors from binding to DNA, resulting in gene repression [16,17]. Catalysis of this process is facilitated by DNA methyltransferases (DNMTs), including DNMT1, DNMT3A, and DNMT3B [18]. Nuclear proteins of the methyl-CpG- binding protein domain influence gene expression by binding to methylated DNA and mobilizing histone deacetylase enzymes, which remodel the chromatin structure, inducing gene silencing [14]. Dysregulated DNA methylation is associated with diseases such as chronic inflammation and cancer, where aberrant methylation can silence tumor-suppressor genes or activate pro-inflammatory genes [19].

### 2.3. Histone Modification Mechanisms

Post-translational modifications (PTMs) of histones (such as acetylation, methylation, phosphorylation, and ubiquitination) regulate chromatin structure and gene expression [20]. These modifications alter nucleosome stability by determining whether a region of chromatin is in a relaxed, transcriptionally active euchromatin state or a condensed, transcriptionally repressive heterochromatin state [21]. Histone acetylation weakens histone–DNA interaction by neutralizing the positive charge on histone tails. This decreased affinity allows for the transcription factors to be accessed, causing gene activation [21]. Collectively, the pattern of histone modifications creates a histone code that is read by the cell’s machinery to organize chromatin dynamics and transcriptional outcomes [22].

## 3. Neutrophil Function and Dysfunction

### 3.1. Normal Role of Neutrophils in the Immune System

Neutrophils are the primary circulating leukocyte in the blood, making up 65% of all circulating leukocytes of the innate immune system [23]. The granule within the cytoplasm contains antimicrobial peptides, enzymes, and ROS, which allows neutrophils to respond to inflammatory stimuli [23]. Through signaling cascades, neutrophils activate NADPH oxidase in response to pathogen-associated molecular patterns (PAMPS) or pathogen phagocytosis [24]. Lactoferrin and metalloproteinases are contained in secondary granules and are associated with inflammatory resolution [25,26], while Arginase-1 (ARG-1), lysozyme, and gelatinase are contained in tertiary granules [27]. ARG-1 metabolizes arginine to ornithine and urea, decreasing the availability of arginine for the generation of nitric oxide (NO) [28].

The release and accumulation of neutrophils in the bone marrow is controlled by the ratio of C-X-C motif chemokine receptor 4 (CXCR4) and granulocyte colony-stimulating factor (G-CSF). CXCR4 expression in neutrophils is retained within the bone marrow, while granulocyte colony-stimulating factor (G-CSF) promotes their migration into the bloodstream [29]. Neutrophil activation and function are modulated by interleukin-23 (IL-23), which enhances the production of interleukin-17 (IL-17) from neutrophils and other immune cells, establishing an important pro-inflammatory feedback loop critical in inflammatory and autoimmune responses [30]. Understanding the mechanisms of neutrophil regulation provides therapeutic targets for ARDS [29].

### 3.2. Dysregulation of Neutrophil Function in ARDS

After contact with the inflammatory insult, neutrophils produce ROS, degranulate proteolytic enzymes, and release NETs. This state of hyperactivation leads to increased permeabilization of the alveolar–capillary barrier [8]. When neutrophils enter pulmonary circulation, the microarchitecture of lung capillaries is smaller than the diameter of the neutrophil, causing these cells to marginate slowly [31]. Inflammatory stimuli stiffen neutrophils through actin polymerization, trapping them in capillaries [32,33]. Neutrophils bind to activated endothelial cells through selectins and integrins, then transmigrate into the interstitial and airspaces [34]. However, neutrophil adhesion itself can disrupt the endothelial barrier since these neutrophils induce cytoskeletal remodeling in endothelial cells, increasing permeability [34]. Neutrophil extravasation then occurs, contributing to the leakage of fluid and proteins into the alveolar space [34,35]. The recruitment of neutrophils into the lungs, combined with their interactions with the endothelium, induces lung injury.

ARDS patients display neutrophil subsets characterized by a long life span and delayed apoptosis, which significantly contributes to a heightened inflammatory state [36]. Additionally, an elevated neutrophil-to-lymphocyte ratio is observed in these patients and aligns with ARDS severity and poor clinical outcomes [37]. These results suggest that the overactivation of neutrophils is a critical factor in the pathogenesis of ARDS. A potential therapeutic option for these patients would be deoxyribonuclease I (DNase I), due to its ability to degrade NETs, mitigating inflammatory damage [38]. In fact, most pulmonary injury is associated with the dysregulation of NETs since their components induce endothelial injury, epithelial cell apoptosis, and microvascular thrombosis [39]. Neutrophil functioning is important in infectious disease to target the disease-causing agent through phagocytosis and the controlled production of ROS [24]. However, in ARDS, they exhibit decreased phagocytic capacity while producing excessive extracellular ROS [27]. Neutrophil dysfunction has significant implications for both the pathophysiology and treatment of ARDS; therefore, DNase activity and neutrophil activation are being explored as therapeutic targets [40]. Altering the excessive inflammatory function of neutrophils without affecting their homeostatic function is a key objective in the development of novel therapeutics for ARDS.

## 4. Neutrophil Epigenetic Biomarkers in ARDS

### 4.1. DNA Methylation

Through recent advancements in research, DNA methylation has been regarded as a regulator of neutrophil function in ARDS. Studies reveal that patients with ARDS have distinct DNA methylation patterns in genes associated with the innate immune system [41]. Neutrophil granule protein genes (Azu1, CTSG, and DEFA4) were hypomethylated in patients with sepsis-induced ARDS [42]. This leads to the overexpression of neutrophil proteolytic and antimicrobial pathways, resulting in lung injury. Disrupted methylation landscapes are found in COVID-19-induced ARDS and are associated with alterations in interferon gene signaling [41]. Hypermethylation of genes regulating apoptotic pathways (ROC1, ZNF789, and H1F0) was observed in patients admitted to the ICU and was associated with a high risk of mortality [41]. These findings reveal that dysregulation of DNA methylation in neutrophils may prolong their survival in severe COVID-19-induced ARDS. However, patients who recovered display hypermethylation of AIM2 and a decrease in neutrophils. Additionally, hypermethylation of caspase genes can exacerbate lung damage due to epigenetic silencing of the apoptotic pathway. This phenomenon is illustrated in the left panel of Figure 1, where the promoter of an apoptotic gene is methylated, resulting in the silencing of the promoter and, ultimately, this pathway, which leads to prolonged survival in the neutrophil population and sustained pulmonary inflammation. Altogether, this suggests that DNA methylations may also be a target for neutrophilic therapeutics.

### 4.2. Histone Modifications

Neutrophils from ARDS patients exhibit changes in chromatin-modifying enzymes, reflecting reprograming of their transcriptional potential [43]. The citrullination of histone H3 on arginine residues, catalyzed by peptidyl arginine deiminase 4 (PAD4), is a hallmark of neutrophil-specific modification [44]. Studies have shown that patients with acute lung injury or septic shock syndrome have increased airway cit-H3, which corresponds to a severe prognosis [44]. Class I/IIb HDACs are required to deacetylate neutrophil histone H3, allowing PAD4 to initiate NET formation [45]. The center panel of Figure 1 illustrates how this net formation is facilitated by HDACs and PAD4, resulting in excessive NET release and exacerbating lung tissue damage. A pharmacological study revealed that the inhibition of HDACs led to hyperacetylation of H3, thereby blocking citrullination. HDAC inhibitors decreased the release of NETs and significantly reduced the levels of pro-inflammatory cytokines in this murine sepsis-induced lung injury model, making HDAC inhibitors a candidate for treatment.

### 4.3. Non-Coding RNAs

Non-coding RNAs (miRNAs and lncRNAs) have been associated as neutrophil regulators in ARDS. This modulation alters gene expression at the post-transcriptional level and can result in the promotion of NET formation or inhibition of apoptotic pathways, leading to prolonged neutrophil survival, as shown in Figure 1. Therefore, the therapeutic utilization of miRNA to target neutrophils in ARDS is a topic of interest. miR-223 is extensively known for functioning in neutrophil differentiation in airway inflammation [46,47,48]. This miRNA is highly expressed in neutrophils and blocks neutrophil-driven inflammation [48]. Studies have shown that inflammasome hyperactivation was associated with low levels of this miR-223 and IL-1β secretion. Additionally, elevated miR-223 revealed protective effects against ALI; this may be due to its ability to suppress many inflammatory pathways, specifically targeting the NLRP3 inflammasome. In support of this, miR-223 KO mice presented increased neutrophil infiltration and worsening alveolar damage following inflammatory stimulation. However, hyperexpression of this miRNA improved the lung function barrier [47]. In the context of ARDS, neutrophils have been shown to actively release miR-223 in microvesicles (MVs), which can be taken up by neighboring lung cells, potentially shaping their inflammatory response. Notably, patients with ARDS show elevated levels of circulating MV-bound miR-223 within the plasma, reflecting protection of lung tissue. In severe ARDS, higher levels of plasma miR-223 have paradoxically been associated with worse outcomes, including fewer ventilator-free days and increased mortality at both 30 and 90 days [48]. This suggests that, while miR-223 may be upregulated as a protective mechanism, its release could reflect overwhelming neutrophil activation rather than effective resolution of inflammation. Knowing this, miR-223 may function as a biomarker of disease severity instead of a direct therapeutic signal. Another study focused on miR-146a, which is regulated through the NF-kB pathway and functions as a negative feedback brake in neutrophil activation [49]. MiR-146a KO murine models showed the development of a pro-inflammatory, senescent neutrophil phenotype that releases NETs, promotes thrombosis, and produces excessive cytokines [49]. This suggests that miR-146a functions in neutrophil homeostasis and decreases neutrophil activation and NETosis. Therefore, this miRNA may be of therapeutic interest in treating ARDS patients who have proliferation of neutrophils with this type of phenotypic characterization. MiR-146a may also be of interest to investigate as a potential biomarker for disease severity.

Challenges arise in patient selection, diagnostics, and prognostics in epigenetic therapy. As a result of the heterogeneity of ARDS, currently, there are no stratification tools that are in routine use to guide therapies. This becomes a hurdle for personalized interventions, since an epigenetic drug may benefit patients with a specific epigenetic profile. At the bedside, ARDS subsets cannot be identified. Through advancements in technology, researchers discovered DNA methylation and miRNA signatures that correlate with patient outcomes in ARDS. Additionally, multi-omics approaches such as epigenomics, transcriptomics, proteomics, and metabolomics are being studied in ARDS to understand the complexity of this syndrome and develop novel diagnostics and therapeutics.

## 5. Neutrophil Diversity and ARDS Heterogeneity

### 5.1. Defining Neutrophil Heterogeneity: HDNs and NDNs

Neutrophils respond to the physiological and pathological conditions they encounter, leading to recent discoveries that these cell types are a heterogeneous population [31,50]. Therefore, characterizing the developmental pathways and function of these subsets is essential for revealing their roles in disease and potential targeted therapeutics [51]. Through density centrifugation, neutrophils can be characterized into groups based on their buyout properties: low-density neutrophils (LDNs) and high-density neutrophils (HDNs) [31,50]. Under homeostatic conditions, HDNs are most abundant, while LDNs comprise 5% of PBMCs [50]. LDNs are associated with immunosuppressive disorders, NET formation, and chronic inflammation [31]. LDNs expand under disease states and exhibit abnormal functions. This subset of neutrophils is categorized as immature or mature (activated) [31,52]. Immature neutrophils possess a heightened pro-inflammatory profile, while mature neutrophils display an increased CXCR4 and decreased CD62L phenotype [45].

### 5.2. Single-Cell Insights into Neutrophil Subsets

Single-cell technologies are being utilized in discovering the heterogeneity of neutrophils in ARDS. Such approaches allow researchers to identify neutrophil subclusters based on surface markers and transcriptomic profiles. Recent studies have used these tools to compare neutrophil phenotypes across ARDS etiologies.

#### Profiles in COVID-19 ARDS vs. Classical ARDS

Due to the COVID-19 pandemic, questions have arisen regarding whether classical ARDS and COVID-19 ARDS are immunologically distinct. In a study completed by Panda et al., plasma was obtained from patients with classical ARDS and exposed to neutrophils extracted from healthy donors. The neutrophilic response led to the release of NETs. However, within the same experiment, plasma from COVID-19-induced ARDS patients was exposed to the same neutrophils and a lack of NETs was observed [53]. Results imply that the heightened NET production in COVID-19 is not attributed to circulating inflammatory molecules within the blood. COVID-19 neutrophils expressed high levels of CDllb, CD11a, and CD66b and a decrease in CD62L and CD107a, which indicates degranulation [53]. The neutrophilic response in COVID-19 patients promotes thrombosis [53,54]. Additional studies have revealed that plasma from patients with severe COVID-19 has increased levels of Cit-H3 and myeloperoxidase-DNA (MPO-DNA) in comparison to patients with classical ARDS [55]. In a prospective observational study, 35% of patients with classical ARDS had a hyperinflammatory phenotype, compared to 20% with COVID-19 [56]. Recent findings suggest that NETs in severe COVID-19 are not solely attributed to elevated systemic cytokine production but intrinsic alterations within neutrophils. Transcriptomic profiling of mature neutrophils from SARS-CoV-2 patients revealed a robust type I interferon (IFN-I) gene signature [57]. This study implicates IFN-I as the priming factor for neutrophil inflammasomes [57]. Increased caspase 1 activation and reduced responsiveness to exogenous priming stimuli were observed in this neutrophil subset. Additionally, immature LDNs, which are involved in spontaneous NET formation, were increased in these patients compared to healthy patients or those with mild COVID-19 infection [57].

Targeting neutrophil heterogeneity as a therapeutic strategy in treating ARDS is of significant interest; however, it still comes with many challenges. These strategies can not only promote suppression of pro-inflammatory, NET-forming subtypes but also need to preserve homeostatic neutrophils that maintain antimicrobial and phagocytic function. Next-generation sequencing technologies, such as single-cell RNA sequencing and spatial transcriptomics, support the development of personalized treatments based on the phenotypic profiles of a patient’s neutrophil populations. While these types of studies hold great promise, they also present significant challenges. The most significant challenge arises from the dynamic plasticity of neutrophils, particularly as the disease environment evolves; we highlight this in Figure 2, where we see how different environmental stressors can alter the dominance of LDN or HDN, how cytokine expression can alter polarization, and the microenvironments created by different causative agents of disease can make the neutrophil landscape appear completely different. Moreover, the presence of overlapping epigenetic markers of relevance between the target population and other cell types heightens the potential of off-target problems. This is especially so when identifying differences between highly similar cell types like myeloid-derived suppressor cells, which share many canonical markers with neutrophils. Not only may there be issues of targeting other cell types but also issues with over-suppression of the immune system, leading to unintended impairment of pathogen clearance ability. While promising, translating findings from single-cell analyses into actionable clinical tools remains complex due to technological limitations, cost, and the need for rapid diagnostic platforms. Thus, while targeting neutrophil heterogeneity holds promise, further research is essential to overcome these limitations and ensure safe and effective therapeutic application.

## 6. Epigenetic Therapeutics in ARDS

Neutrophil recruitment and NET formation are key drivers in the pathogenesis of ARDS, and targeting epigenetic pathways involved in these processes may promote lung tissue resolution. Classes of epigenetic modulators such as HDAC inhibitors, PAD4 inhibitors, DNA methyltransferase (DNMT) inhibitors, and protease-activated receptor-1 (PAR1) antagonists have been effective in preclinical models of ARDS, ALI, COVID-19, and related inflammatory conditions; additionally, some therapeutic agents are already in clinical use for other diseases; these therapeutic agents and their targets are outlined in Table 1 [45].

Pharmacological inhibitors of PAD4 include GSK484, JBI-589, GSK199, and BMS-P5. In preclinical ALI models, mice treated with GSK484 exhibited a decrease in dsDNA in the BALF, which is indicative of decreased NETosis [58], in these GSK484-treated mice,

Cit-H3, von Willebrand factor (vWf), and PAD4 expression were also reduced. However, another study exploring the efficacy of GSK484 in SARS-CoV-2-induced lung injury revealed that this inhibitor impacted normal T cell functions [59]. This inhibitor effectively reduced NETs [58,59] but impaired the adaptive immunity. Therefore, GSK484 could be a good option for mitigating neutrophil-associated damage. However, the impact on adaptive immunity may have opposing effects, leading to prolonged disease duration and damage. These alterations also increase the risk of secondary infections due to the compromised immune system. If this inhibitor were to be considered for treatment, it would likely need to be used in combination therapy, where a T cell stimulator would also be required. JB1-589, has been studied in the context of rheumatoid arthritis, lung cancer, and colon cancer [60,61]. In murine cancer experiments, high expression of PAD4 activity and CXCR2 is observed in neutrophils of tumor growths and in the peripheral blood of lung and colon cancer patients [60]. Tumor-bearing mice treated with JB1-598, had reduced lung metastases and tumor growth, accompanied by reduced expression of PAD4 and CXCR2 [60]. Therefore, this inhibitor has the potential to be a therapeutic agent in neutrophil-driven inflammatory diseases.

HDAC inhibitors enhance histone acetylation and chromatin decondensation, leading to the dysregulation of pro-inflammatory genes. Butyrate, an HDAC inhibitor, reduced neutrophil infiltration and decreased production of NO, IL-1β, and TNF- α in the BALF of mice exposed to LPS-induced lung injury [62]. Studies in mice models have shown that HDAC inhibitors reduce pulmonary inflammation and improve survival in sepsis-induced lung injury [45]. Currently, there are no approved HDAC inhibitors specifically for ARDS; agents such as vorinostat, panobinostat, and valproic acid are FDA-approved for other indications, offering potential for drug repurposing in the context of severe lung inflammation [45]. Repurposing HDAC inhibitors for ARDS treatment is of great interest due to their ability to suppress pro-inflammatory gene expression, reduce NET release, and the promising results in preclinical mouse models, as shown in Figure 3. However, histone modifications are broad regulators of gene expression, and this may elicit off-target effects. This would be broad immune suppression that may result in susceptibility to secondary infections. Repurposing drugs in general also comes with additional hurdles of optimizing dosing, assessing adverse effects, and the overall lack of current clinical trials.

Protease-activated receptor 1 (PAR1) is a mediator of angiogenesis and inflammation and regulates neutrophil recruitment [63]. In murine LPS-induced lung injury, PAR1 antagonists decreased neutrophil accumulation in the lung and preserved the alveolar capillary [64]. This therapeutic approach is attributed to the decrease in CCL2 and CCL7, which are involved in neutrophil chemotaxis [64]. These results reveal that PAR1 is a promising therapeutic target in ARDS, and its antagonism has the potential to reduce immune cell infiltration in the lungs [64]. DNMT inhibitors, such as 5-azacytidine (Aza), promote neutrophil apoptosis through activation of the death-associated protein kinase 1 (DAPK1) [65]. Murine lung injury models administering Aza led to a downregulation of proteins associated with anti-apoptotic proteins through the NF–NF-κβ signaling pathway. This led to enhanced inflammatory resolution through neutrophil clearance and decreased production of pro-inflammatory cytokines [65]. Preclinical studies demonstrated that Aza effectively suppressed pulmonary edema and decreased neutrophil infiltration in bronchoalveolar lavage fluid.

Understanding the function of epigenetic modification in the context of ARDS opens opportunities for the utilization of epigenetic modulators, such as PAD4 inhibitors, HDAC inhibitors, DNMT inhibitors, and PAR1 antagonists, for precision treatment. While several of these agents are FDA-approved for the treatment of other diseases, new interest is focused on repurposing these agents for treatment in ARDS. Aside from using these compounds as treatments, we can start to investigate their targets as biomarkers of disease prognosis as well. Therefore, integrating these strategies to target neutrophil-driven pathology and identify patient risk will help improve patient outcomes in this fatal disease but only after extensive research evaluates the efficiency and broader immune impacts.

## 7. Limitations of Current Studies and Conclusions

The discrepancies in understanding between the clinical manifestations of ARDS and the experimental models designed to investigate these consequences are frequently linked to numerous constraints. These constraints are succinctly depicted in Figure 4, which highlights numerous variables, ranging from expense to insufficient understanding of ethnic variations in human research, all of which increases the difficulty with targeting neutrophil epigenetics for therapeutic strategies and biomarker identification. This text broadly assesses some constraints and examines them within individual investigations.

The study of neutrophil function through in vitro experiments (HL-60 cell line) does not wholly recapitulate neutrophil epigenetics. These cells are beneficial when studying the mechanism of apoptotic pathways in neutrophils; however, they are not representative of the behavior of primary neutrophils. The use of primary human neutrophils poses challenges due to their fragility and short lifespan, which makes it difficult to employ them in epigenetic studies. Alternatively, murine models have demonstrated significant utility in elucidating the systemic neutrophilic response; fewer heterogenous populations of neutrophils are observed in murine models compared to humans. A total of 50–70% of WBCs are neutrophils for humans, while they make up 10–30% in mice [66]. Therefore, mice may not display the full spectrum of neutrophil subsets seen in human ARDS cases. To induce ARDS in animals, most studies use a single mechanism such as a lipopolysaccharide (LPS), staphylococcus enterotoxin B (SEB), hydrochloric acid (HCL), or bleomycin. Human ARDS can involve multiple factors such as having pneumonia and being placed on a ventilator. Comorbidities such as shock, blood transfusions, sepsis, pancreatitis, and chronic diseases may induce epigenetic changes that are not even seen in some simplified models of ARDS, which ultimately limit their prognostic and diagnostic potential. These comorbidities are often overlooked in preclinical murine models.

Epigenetic regulation of neutrophils in ARDS has revealed that neutrophils have increased plasticity compared to that previously perceived, which contributes to the pathogenesis of this disease. However, contradictory observations in the literature reveal that these changes in neutrophil subsets are context-dependent. Modulation of the epigenetic switches that control neutrophils may target the inflammatory response in ARDS and preserve host defenses. Nonetheless, off-target responses may result in unregulated immunosuppression and heightened susceptibility to subsequent infections. Additional aspects to examine include age, ethnicity, and biological sex, which influence neutrophil biology and must be considered in the development of targeted therapies for ARDS. For example, research has demonstrated that young adult females exhibit a mature and activated neutrophil phenotype in comparison to their male counterparts of the same age [67]. These disparities can be ascribed to sex-specific epigenetic modulation of immune genes, leading to divergent treatment results among patients in these demographics. Other reports focused on SARS-CoV-2-induced ARDS have demonstrated that males have a higher mortality compared to females [68,69]. These differences could be attributed to epigenetic changes or hormonal influences systemically, which would need to be further explored before embarking on our proposed therapeutic strategies. Specifically, estrogen has been shown to exhibit anti-inflammatory and immunomodulatory effects and, in SARS-CoV-2-induced ARDS, it downregulates angiotensin-converting enzyme 2 (ACE2), hence preventing viral entry into the cell [70]. Human cell line studies further investigating the impact of hormonal regulation of neutrophil activity in ARDS revealed that estradiol can mediate suppression of NET formation in male neutrophil cell lines, while testosterone enhanced NETosis in female neutrophils [71]. This study did not examine the epigenetic mechanisms underlying these reactions; however, these responses should be taken into account when determining alternative therapies and represent an additional limitation of this research. Aside from sex-specific responses that are being explored, another limitation of previously conducted preclinical research and clinical trials is the representation and understanding of neutrophilic and epigenetic variability in minority populations. Clinical trials generally exhibit inadequate minority participation, a trend also evident in ARDS research, where meta-analysis indicated that merely 30.4% of participants in an ARDS clinical trial were from Black, Hispanic, or other non-white backgrounds altogether [72]. This is crucial, as the phenotypes of neutrophils and DNA methylation patterns may differ based on an individual’s heritage. Thus, biomarkers utilized for diagnosing ARDS in one cohort may not be relevant in another. Benign ethnic neutropenia (BEN) is a disease characterized by reduced neutrophil counts, predominantly reported in individuals of African heritage [73]. This indicates substantial differences that may not be addressed in preclinical studies, which may lead to diverse responses in clinical applications. Additional limitations to consider in the evaluation of epigenetic regulations of neutrophils for targeted therapies or biomarkers include the timing of interventions and the costs associated with precision medicine. A comprehensive understanding of neutrophil epigenetics and heterogeneity may lead to innovative, targeted therapies that promote resolution and address the underlying immune dysregulation in ARDS, thereby enhancing patient outcomes. However, many limitations must be addressed.

## Figures and Tables

**Figure 1 cells-14-01151-f001:**
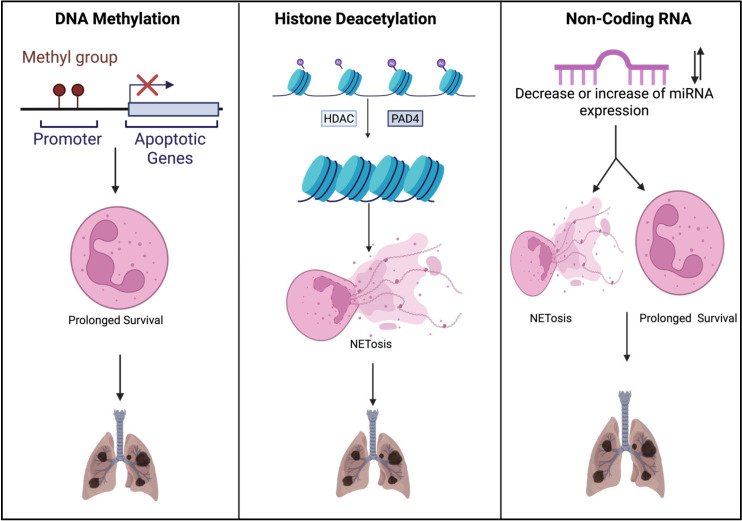
Epigenetic regulation of neutrophils in ARDS. This figure illustrates how DNA methylation, histone modifications, and non-coding RNAs contribute to neutrophil-induced lung injury in ARDS.

**Figure 2 cells-14-01151-f002:**
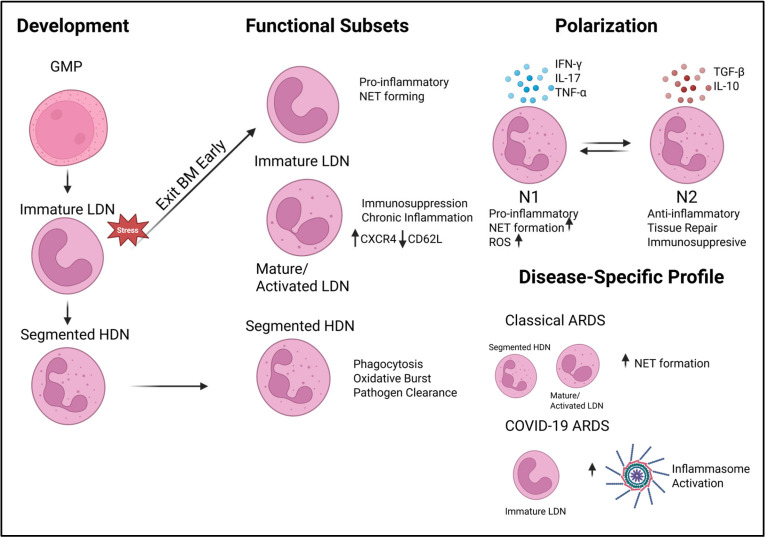
Neutrophil heterogeneity depicted in development and in ARDS. This schematic illustrates neutrophil development from granulocyte-monocyte progenitors (GMPs) and the stress-induced release of immature low-density neutrophils (LDNs), as well as fully mature segmented high-density neutrophils (HDNs). Functional plasticity is demonstrated by the polarization of cells into N1 (pro-inflammatory) and N2 (regulatory) phenotypes, which are influenced by environmental cytokines. Disease-specific profiles highlight the expansion of immature LDNs and increased NET formation in ARDS.

**Figure 3 cells-14-01151-f003:**
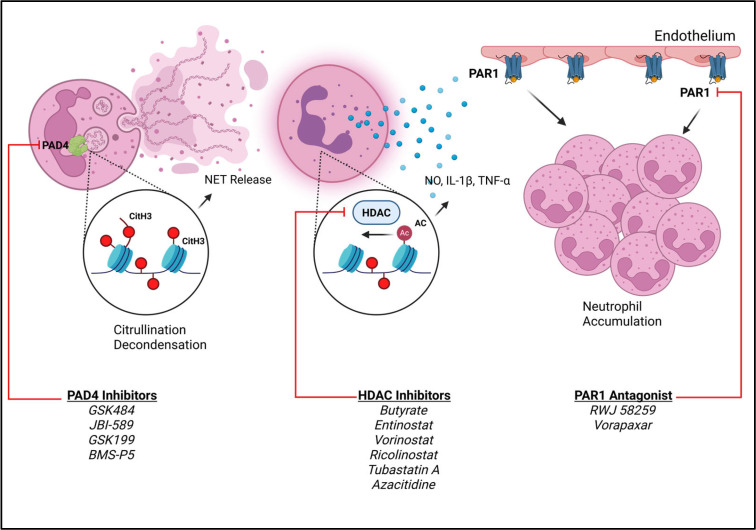
Mechanisms of epigenetic inhibitors in neutrophil activation and recruitment in ARDS. This figure illustrates key epigenetic mechanisms involved in neutrophil NET formation, activation, and recruitment via regulation by PAD4, HDAC, and PAR1. Inhibitors or antagonists are highlighted here that directly target each of the described epigenetic modulators.

**Figure 4 cells-14-01151-f004:**
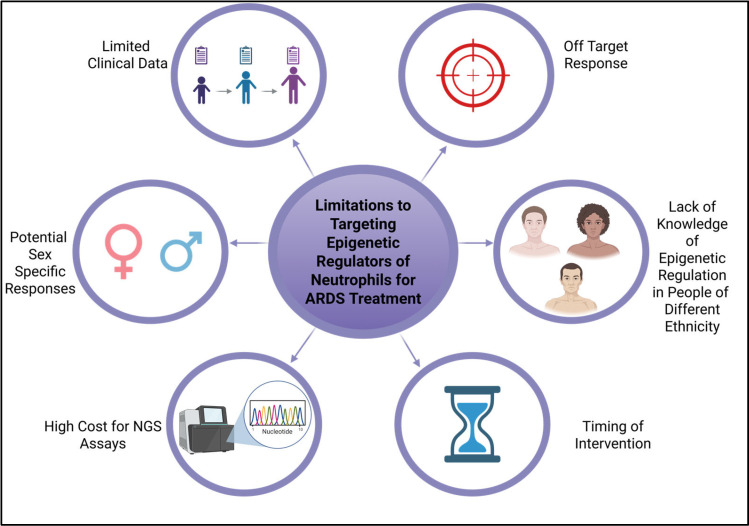
Limitations to targeting epigenetic regulators of neutrophils for the treatment of ARDS. This schematic highlights key challenges associated with using epigenetic therapies to modulate neutrophil function in ARDS, including patient heterogeneity (age, sex, and ethnicity), non-specificity of available inhibitors, timing of intervention, lack of biomarker-guided precision strategies, dynamic epigenetic landscapes, and variable treatment responses across populations.

**Table 1 cells-14-01151-t001:** Epigenetic modulators of neutrophil activity: therapeutic targets and inhibitors relevant to ARDS.

Target	^a^ Preclinical Evidence Strength	^b^ Clinical Development Status	^c^ Role in Neutrophil Biology	^d^ Example Inhibitor Compounds
PAD4	Strong	Preclinical	Drives NETosis through histone citrullination	JBI-589, GSK199, GSK484, BMS-P5
HDAC 1/2/3	Strong	Clinical (Oncology)	Pro-inflammatory gene expression and NETosis	Entinostat, Vorinostat
HDAC 6	Strong	Clinical (Oncology)	NET formation	Ricolinostat, TubastatinA
DNMT3A	Moderate	Approved (Oncology)	Loss of methylation leads to inflammation	Azacitidine

^a^ Preclinical Evidence Strength refers to the level of experimental support from both in vitro and preclinical animal studies evaluating the target’s role in neutrophil regulation. ^b^ Clinical Development Status indicates whether the inhibitors are in clinical trials or approved for other disease indications, primarily oncology at present. No inhibitors listed are approved for ARDS treatment. ^c^ Role in Neutrophil Biology describes the specific role the target plays in neutrophil biology relevant to ARDS pathogenesis. ^d^ Example Inhibitor Compounds lists representative drugs targeting each epigenetic modulator.

## Data Availability

No new data were created or analyzed in this study. Data sharing is not applicable to this article.

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
