# Peer review of "Epigenetic Regulation of Neutrophils in ARDS"

_cells, 2025, doi:10.3390/cells14151151_

Round 1
Reviewer 1 Report
Comments and Suggestions for Authors
- While the topic is relevant, true novelty or synthesis is limited. The review mostly catalogs known findings without introducing a new conceptual framework, model, or hypothesis that might unify these mechanisms. Please introduce a conceptual figure that illustrates how different epigenetic changes (e.g., methylation, acetylation, ncRNAs) integrate to control neutrophil fate and function.
- There is no critical comparison of studies (e.g., contradictory findings, methodological differences) or discussion of knowledge gaps. Therefore, please include a critical perspective on limitations of current models and outstanding questions.
- The Introduction is too long and contains significant redundancy. The background on ARDS and neutrophils could be condensed to avoid overwhelming the reader before reaching the main epigenetic focus. Please merge Sections 1.1–1.3 into a single streamlined section (~1 page max) summarizing ARDS and neutrophils, to keep the focus on the review’s core topic.
- Some mechanistic explanations are repetitive or superficial, especially in the PAD4 and HDAC sections.
- The review conflates cell-type specific effects; for example, it is not always clear whether findings refer to neutrophils, monocytes, or epithelial cells. Please clarify which findings are specific to neutrophils versus other lung-resident or infiltrating immune cells.
- More careful source attribution is needed when generalizing from in vitro or animal studies. Please briefly address technical limitations (e.g., lack of neutrophil-specific epigenomic profiling tools due to short lifespan and limited transcriptional activity).
- Inconsistent or incorrect punctuation, grammar, and spacing appear throughout the manuscript (e.g., “decondensation and NET release. Limits neutrophil-driven” is a sentence fragment).
- There are several spelling and formatting errors (e.g., "a enuate", "chemoattractant" spelled as "chemoat- tractant", etc.).
- Inconsistent citation style: some references are repeated; others are misnumbered or misspelled.
- The manuscript lacks any figures or diagrams, which are crucial in review, especially to visualize pathways, feedback loops, and therapeutic strategies. Please add i) graphical summary of epigenetic mechanisms in neutrophils during ARDS, ii) a conceptual model showing the feedback between cytokines, NETs, and epigenetic modulators and iii) timeline or flow chart of neutrophil activation → dysfunction → resolution.
- References: Some critical recent studies on single-cell or multi-omics analysis in ARDS, and epigenetic landscape mapping in neutrophils, are missing. Please ensure that all claims are properly sourced and unique studies are not cited multiple times under different numbers. Moreover, consider citing key reviews from Nature Reviews Immunology, Immunity, or Cell Reports Medicine for broader context.
Author Response
Comment: 1 While the topic is relevant, true novelty or synthesis is limited. The review mostly catalogs known findings without introducing a new conceptual framework, model, or hypothesis that might unify these mechanisms. Please introduce a conceptual figure that illustrates how different epigenetic changes (e.g., methylation, acetylation, ncRNAs) integrate to control neutrophil fate and function.
Response 1: We added section 4 entitled Neutrophil Epigenetic Biomarkers in ARDS and included a summary schematic of how these epigenetic changes impact neutrophil fate and function.
Comment 2: There is no critical comparison of studies (e.g., contradictory findings, methodological differences) or discussion of knowledge gaps. Therefore, please include a critical perspective on limitations of current models and outstanding questions.
Response 2: We expanded on our conclusion in section 7 and addressed limitations in ARDS studies.
Comment 3: The Introduction is too long and contains significant redundancy. The background on ARDS and neutrophils could be condensed to avoid overwhelming the reader before reaching the main epigenetic focus. Please merge Sections 1.1–1.3 into a single streamlined section (~1 page max) summarizing ARDS and neutrophils, to keep the focus on the review’s core topic.
Response 3: Thank You. We condensed sections 1.1-1.3
Comment 4: Some mechanistic explanations are repetitive or superficial, especially in the PAD4 and HDAC sections.
Response 4: We removed some of the repetitive information regarding PAD4 and HDACs.
Comment 5: The review conflates cell-type specific effects; for example, it is not always clear whether findings refer to neutrophils, monocytes, or epithelial cells. Please clarify which findings are specific to neutrophils versus other lung-resident or infiltrating immune cells.
Response 5: Added section that that focuses on Neutrophil Diversity and Heterogeneity
Comment 6: More careful source attribution is needed when generalizing from in vitro or animal studies. Please briefly address technical limitations (e.g., lack of neutrophil-specific epigenomic profiling tools due to short lifespan and limited transcriptional activity).
Response 6: Added to section 7 in Limitations and Conclusion
Comment 7: Inconsistent or incorrect punctuation, grammar, and spacing appear throughout the manuscript (e.g., “decondensation and NET release. Limits neutrophil-driven” is a sentence fragment).
Response 7: Thank you. We went back through to correct grammatical errors
Comment 8: There are several spelling and formatting errors (e.g., "a enuate", "chemoattractant" spelled as "chemoat- tractant", etc.).
Response 8: Thank you. We went back through to correct grammatical errors
Comment 9: Inconsistent citation style: some references are repeated; others are misnumbered or misspelled.
Response 9: We went back and looked over the reference list.
Comment 10: The manuscript lacks any figures or diagrams, which are crucial in review, especially to visualize pathways, feedback loops, and therapeutic strategies. Please add i) graphical summary of epigenetic mechanisms in neutrophils during ARDS, ii) a conceptual model showing the feedback between cytokines, NETs, and epigenetic modulators and iii) timeline or flow chart of neutrophil activation → dysfunction → resolution.
Response 10: We added a graphical abstract and included a schematic in section 4 entitled Neutrophil Epigenetic Biomarkers in ARDS
Comment 11: References: Some critical recent studies on single-cell or multi-omics analysis in ARDS, and epigenetic landscape mapping in neutrophils, are missing. Please ensure that all claims are properly sourced and unique studies are not cited multiple times under different numbers. Moreover, consider citing key reviews from Nature Reviews Immunology, Immunity, or Cell Reports Medicine for broader context.
Response 11: Added section 5 focused on Neutrophil Diversity and ARDS Heterogeneity.
Reviewer 2 Report
Comments and Suggestions for Authors
This timely review synthesizes cutting-edge insights into epigenetic regulation of neutrophils in ARDS pathogenesis, highlighting promising therapeutic avenues. The focus on NETosis, histone modifications, and non-coding RNAs aligns with emerging trends in immunology and offers translational relevance. This an interesting manuscript, but I have several following concerns:
- There are limited discussion of human ARDS heterogeneity (e.g., COVID-19 vs. sepsis-induced ARDS). Please address how neutrophil epigenetics may differ across ARDS etiologies.
-
There are over 20 epigenetic regulators are implicated in neutrophils; the review should highlight the most therapeutically promising (e.g., PAD4 for NETosis suppression). Please add a table ranking targets by:
-
Strength of preclinical evidence
-
Drug development status
-
Potential for rapid clinical translation
-
-
Neutrophil-specific targeting remains a hurdle (systemic epigenetic inhibitors risk off-tissue effects). Please discuss novel delivery systems (e.g., nanoparticle-encapsulated PAD4 inhibitors for lung targeting).
-
Epigenetic marks (e.g., NET-associated DNA methylation) could serve as ARDS diagnostic/prognostic tools. Please dedicate a subsection to epigenetic biomarkers in ARDS stratification.
-
Pathogen-Specific Nuances: How do bacterial vs. viral ARDS alter neutrophil epigenetics?
- Can epigenetic reprogramming prevent ARDS in high-risk patients? Please add some discussions about these concerns.
- The tables in the text should use a standard three-line form.
- Ensure that all acronyms (e.g., ARDS in Line 2, "CXCR4" in Line 175) are clearly defined at first mention in the abstract and the main text if not already done. Please double check all the text and correct them.
-
Please scheme "Epigenetic Control Points in ARDS Neutrophils" with:
-
Histone modifications → Cytokine release
-
PAD4 → NETosis → Tissue damage
-
miRNAs → Inflammatory gene silencing
-
- Please unify the format of references in the article, including the author's name, the case of words in the title of the article, the writing of the name of the journal, and the page number.
- The repetition rate of the article is as high as 33%, in order to avoid the suspicion of plagiarism, please reduce the repetition rate of the full text.
The English could be improved to more clearly express the research.
Author Response
Comment 1: There are limited discussion of human ARDS heterogeneity (e.g., COVID-19 vs. sepsis-induced ARDS). Please address how neutrophil epigenetics may differ across ARDS etiologies.
Response 1: Added Section 5 entitles Neutrophil Diversity and ARDS Heterogeneity
Comment 2: There are over 20 epigenetic regulators are implicated in neutrophils; the review should highlight the most therapeutically promising (e.g., PAD4 for NETosis suppression). Please add a table ranking targets by:
- Strength of preclinical evidence
- Drug development status
- Potential for rapid clinical translation
Response 2: Added table to section 6: Epigenetic Therapeutics in ARDS.
Comment 3: Neutrophil-specific targeting remains a hurdle (systemic epigenetic inhibitors risk off-tissue effects). Please discuss novel delivery systems (e.g., nanoparticle-encapsulated PAD4 inhibitors for lung targeting).
Response 3: Mentioned off target effects in Section 6
Comment 4: Epigenetic marks (e.g., NET-associated DNA methylation) could serve as ARDS diagnostic/prognostic tools. Please dedicate a subsection to epigenetic biomarkers in ARDS stratification.
Response 4: Added section 4: Neutrophil epigenetic biomarkers in ARDS
Comment 5: Pathogen-Specific Nuances: How do bacterial vs. viral ARDS alter neutrophil epigenetics?
Response 5: Added Section 5.2.1 Profiles in COVID-19 ARDS vs classical ARDS
Comment 6: Can epigenetic reprogramming prevent ARDS in high-risk patients? Please add some discussions about these concerns.
Response 6: In section 4: Neutrophil Epigenetic Biomarkers in ARDS, epigenetic differences in patients with and without ARDS are discussed.
Comment 7: The tables in the text should use a standard three-line form.
Response 7: Revised my table in section 6
Comment 8: Ensure that all acronyms (e.g., ARDS in Line 2, "CXCR4" in Line 175) are clearly defined at first mention in the abstract and the main text if not already done. Please double check all the text and correct them.
Response 8: Checked the acronyms to ensure they are all clearly defined
Comment 9: Please scheme "Epigenetic Control Points in ARDS Neutrophils" with:
- Histone modifications → Cytokine release
- PAD4 → NETosis → Tissue damage
- miRNAs → Inflammatory gene silencing
Response 9: Added figure 1 (Epigenetic mechanisms of Neutrophils in ARDS) to section 4
Comment 10: Please unify the format of references in the article, including the author's name, the case of words in the title of the article, the writing of the name of the journal, and the page number.
Response 10: Revised the references
Comment 11: The repetition rate of the article is as high as 33%, in order to avoid the suspicion of plagiarism, please reduce the repetition rate of the full text.
Comment 11: Revised some of the sections
Round 2
Reviewer 1 Report
Comments and Suggestions for Authors
1. The authors did reply to my comments superficially and did not indicate in their manuscript where they did changes.
2. A good review should have at least 4 figures.
3. It is still not clear what new conception the review is providing.
4. While the manuscript gathers a large amount of information, it often reads like a collection of factsrather than a critical synthesis. Review articles should not only report but integrate and evaluate For example, in Section 4.3, different studies on miR-223 are listed without reconciling contradictions or discussing their context.
5. There are numerous grammatical, typographic, and stylistic errors, which compromise readability and professionalism.
6. Some concepts are repeated unnecessarilyacross sections (e.g., NETs formation, DNase I activity, role of miR-223).
7. Figure 1is referred to but not described in the text in any detail. The caption is also poorly written ("in ARRDS").
8. Table 1 lacks clear formatting and explanation. The columns are confusing.
9. The limitations section is a good start but should be expanded. Some key missing elements, such as sex/gender differences, ethnicity, or clinical trial hurdles in applying epigenetic therapies.
10. “miR-233” is used throughout but should likely be miR-223.
11. Inconsistent referencing such as numbers in brackets not always corresponding to final reference list.
Author Response
- The authors did reply to my comments superficially and did not indicate in their manuscript where they did changes.
We apologize for the brief response to the previous comments. To remedy this issue, we have tracked changes and shared these tracked changes in a separate document. Additionally, we have highlighted regions in which new material was added.
- A good review should have at least 4 figures.
Thank you for bringing this to our attention. We have now included 3 additional figures to increase readability and manuscript flow.
- It is still not clear what new conception the review is providing.
The concept for this review is to introduce the idea of using epigenetic regulators of neutrophils as predictive biomarkers of disease severity and targets for precision therapeutics. We emphasized this throughout the manuscript. All edits are highlighted in the tracked revisions file.
- While the manuscript gathers a large amount of information, it often reads like a collection of facts rather than a critical synthesis. Review articles should not only report but integrate and evaluate For example, in Section 4.3, different studies on miR-223 are listed without reconciling contradictions or discussing their context.
To emphasize the purpose of the article, we have re-evaluated these studies and applied critical thoughts to these studies and their applicability to our greater concepts. All edits are highlighted in the tracked revisions file.
- There are numerous grammatical, typographic, and stylistic errors, which compromise readability and professionalism.
Our apologies for these extensive errors. As a result, we have thoroughly reviewed the article to mitigate these issues to the best of our knowledge. All edits are highlighted in the tracked revisions file.
- Some concepts are repeated unnecessarily across sections (e.g., NETs formation, DNase I activity, role of miR-223).
We have reduced the redundancy of concepts as much as possible while maintaining a proper understanding of content.
- Figure 1 is referred to but not described in the text in any detail. The caption is also poorly written ("in ARRDS").
The caption for Figure 1 has been revised, including the spelling error for ARDS. Additionally, this figure has now been cited in multiple locations, providing descriptions on the mechanisms illustrated in each panel. All edits are highlighted in the tracked revisions file.
- Table 1 lacks clear formatting and explanation. The columns are confusing.
Table 1 has been updated to increase clarity by updating the table title along with the column titles. Additionally, we have also included footnotes to describe what we are evaluating in each column.
- The limitations section is a good start but should be expanded. Some key missing elements, such as sex/gender differences, ethnicity, or clinical trial hurdles in applying epigenetic therapies.
As suggested, we have expanded this section. All edits are highlighted in the tracked revisions file.
- “miR-233” is used throughout but should likely be miR-223.
Thank you for bringing this to our attention. We have made this correction throughout the article to ensure that this typo is corrected.
- Inconsistent referencing such as numbers in brackets not always corresponding to final reference list.
Our apologies for the mistake. We have double checked endnote and adjusted references to ensure the in-text citations are in order, relevant, and reflected properly in the reference section.
Reviewer 2 Report
Comments and Suggestions for Authors
The authors have addressed all my comments, I recommend accepting it in curretn form.
Author Response
No comments to respond to at this time.
Round 3
Reviewer 1 Report
Comments and Suggestions for Authors
none